# Observation of the Ionosphere in Middle Latitudes during 2009, 2018 and 2018/2019 Sudden Stratospheric Warming Events

**Zbyšek Mošna** [1,*] **, Ilya Edemskiy** [1,2] **, Jan Laštovička** [1] **, Michal Kozubek** [1] **, Petra Koucká Knížová** [1] **, Daniel Kouba** [1] **and Tarique Adnan Siddiqui** [3]

1   Department of Ionosphere and Aeronomy, Institute of Atmospheric Physics, Czech Academy of Sciences, 141 31 Prague, Czech Republic; ilya@iszf.irk.ru (I.E.); jla@ufa.cas.cz (J.L.); kozubek.michal@ufa.cas.cz (M.K.); pkn@ufa.cas.cz (P.K.K.); kouba@ufa.cas.cz (D.K.)
2   Institute of Solar-Terrestrial Physics, Russian Academy of Sciences, 664033 Irkutsk, Russia
3   Leibniz-Institute of Atmospheric Physics, 18225 Kühlungsborn, Germany; siddiqui@iap-kborn.de
*   Correspondence: zbn@ufa.cas.cz

**Abstract:** The ionospheric weather is affected not only from above by the Sun but also from below by processes in the lower-lying atmospheric layers. One of the most pronounced atmospheric phenomena is the sudden stratospheric warming (SSW). Three major SSW events from the periods of very low solar activity during January 2009, February 2018, and December 2018/January 2019 were studied to evaluate this effect of the neutral atmosphere on the thermosphere and the ionosphere. The main question is to what extent the ionosphere responds to the SSW events with focus on middle latitudes over Europe. The source of the ionospheric data was ground-based measurements by Digisondes, and the total electron content (TEC). In all three events, the ionospheric response was demonstrated as an increase in electron density around the peak height of the F2 region, in TEC, and presence of wave activity. We presume that neutral atmosphere forcing and geomagnetic activity contributed differently in individual events. The ionospheric response during SSW 2009 was predominantly influenced by the neutral lower atmosphere. The ionospheric changes observed during 2018 and 2018/2019 SSWs are a combination of both geomagnetic and SSW forcing. The ionospheric response to geomagnetic forcing was noticeably lower during time intervals outside of SSWs.

**Keywords:** sudden stratospheric warmings; ionospheric effects; ionospheric variability; vertical coupling

## 1. Introduction

The Earth's ionosphere is created by solar-ionizing EUV and X-ray radiation and energetic particle precipitation.

The term space weather refers to conditions on the Sun and in the solar wind, magnetosphere, ionosphere, and thermosphere that can influence the performance and reliability of space-borne and ground-based technological systems and that can affect human life and health (definition used by the U.S. National Space Weather Plan [1]). The ionosphere is a part of the atmosphere which significantly contributes to the propagation of radio waves, and therefore, it influences the quality of global navigation satellite systems (GNSS) and other technologies.

Besides the solar and geomagnetic forcing, the ionosphere is modulated by processes in the neutral atmosphere, which contribute to the ionospheric part of space weather (ionospheric weather). Important factors are processes initiated in the lower atmosphere (troposphere and stratosphere) as various upward propagating atmospheric waves (planetary, tidal, gravity, and infrasonic; e.g., [2]) and the sudden stratospheric warmings (SSWs, e.g., [3,4]). The latter occur in the wintertime high-latitude stratosphere, essentially at the Northern Hemisphere. There are several types of SSWs, namely major, minor, Canadian, or

final [5] according to presence or absence of the high-latitude zonal wind reversal (ZWR) at latitude of 60°. Detailed review of SSWs is given in [6].

Here we focus on effects of major SSW on the main ionospheric parameters, the critical frequency of F2 layer (foF2), the height of F2 layer maximum (hmF2), the critical frequency of E-layer (foE), the height of E-layer (hmE), electron density profiles, and the observed and modeled total electron content (TEC). There are also specific effects in the lower ionosphere below 100 km, which are of different morphology and nature (e.g., [7] and references herein), but these effects are out of the scope of this paper.

Ionospheric effects of SSWs have been relatively intensively studied in the last decade, particularly those of the January 2009 SSW, because unexpected and strong effects of this SSW had been observed in the low-latitude ionosphere (e.g., [8–10]). Results from the first period of investigations of ionospheric effects of SSWs were reviewed by [11]. Ionospheric effects of SSWs at low latitudes are longitudinally dependent [12,13]. The effects of Arctic SSWs were observed also in the southern low-latitude ionosphere [14–16]. A strong thermospheric cooling accompanied the January 2009 SSW [17], which is a feature of typical temperature response to a major SSW. The equatorial ionosphere response to SSW is distinctly different for different phases of the quasi-biennial oscillation (QBO) [18,19]. A strongly enhanced lunar semidiurnal tide plays an important role in ionospheric effects of SSWs at low latitudes [8,20]. Simultaneous analysis of temperatures in the stratosphere-mesosphere-lower-thermosphere and TEC during SSW 2013 reveals somewhat changing spectral content of tides with altitude, possibly due to nonlinear interactions with planetary waves [21]. Numerical simulations confirm important role of changes in the migrating semidiurnal solar (SW2) and lunar (M2) tides as well as in the westward propagating nonmigrating semidiurnal tide with zonal wavenumber 1 (SW1) [22]. Based on the thermosphere-ionosphere-electrodynamics general circulation model (TIE-GCM) simulations, it has been shown that the major SSW forcing is a significant factor strongly modifying the effect of major geomagnetic storm in equatorial ionosphere by up to 100% of storm-induced TEC change [23]. Model EAGLE (entire atmosphere global model) shows that the phase change of SW2 in the neutral wind caused by the 2009 SSW at the altitude of the dynamo electric field generation had a crucial importance for the observed low-latitudinal TEC disturbances [24]. The equatorial electrojet plays a key role in SSW-induced changes of TEC in low latitudes but not at middle latitudes [25,26]. Effects of the September 2019 southern SSW have been observed at low latitudes in the topside ionosphere by Swarm satellites by [27] and in TEC by [28]. The effects of SSWs have also been studied in the midlatitudinal ionosphere (e.g., [13,29–31]) but much less than at low and equatorial latitudes. Also minor SSWs, not only major SSWs, are capable of significant modification of the midlatitude ionosphere [32]. In the American sector, the nighttime SSW-induced TEC perturbations in ~55° S–45° N were found to be negative and substantially stronger than daytime perturbations [33]. Both the Constellation Observing System for Meteorology, Ionosphere, and Climate (COSMIC) observations and Thermosphere Ionosphere Mesosphere Electrodynamics General Circulation Model (TIME-GCM) simulations reveal perturbations in hmF2 at the Southern Hemisphere midlatitudes during SSW 2009 and 2013 time periods, which are ~20–30 km which correspond to 10–20% variability of the background mean hmF2 [34]. The high latitude Arctic ionosphere reveals signatures of SSWs as well [35]. Decrease in foF2 Digisonde derived parameters for Irkutsk and Yakutsk station for the 2009 SSW was shown in [36].

The authors [37] observed foF2 and hmF2 and electron density data at fixed heights. They observed decrease in foF2 and hmF2 on 16 day averaged data in middle and low altitudes. The proposed mechanism is similar to the so-called "disturbed dynamo", in this case caused by winds originating at high latitudes due to thermospheric heating. Due to conservation of angular momentum, the wind is continuing westward which results in downward (or less positive) vertical drift leading to increase in recombination and decrease in electron density.

Analysis of ionospheric response to the 2008 minor SSW using Irkutsk, Kaliningrad, Sao Jose dos Campos, and Jicamarca data was performed by [25]. The decrease in F2 region electron density was explained as change in ratio between O and $N_2$ as well as change in zonal electric field. They observed decrease in foF2 on days corresponding to maximal positive disturbance in stratospheric temperature

Well-seen gradual decrease in midday TEC maximum from 19 to 24 January 2009 for Irkutsk, Novosibirsk (inside the stratospheric cyclone), and Yakutsk stations was observed during midday [30]. Daytime decrease in NmF2 values at 10–20% during the 2006–2013 SSW starts and maxima compared to background level at Norilsk arctic station was reported. After the SSW maxima, the opposite increase in NmF2 was observed lasting 10–20 days [35]. As discussed in the paper of [38], dealing with the high-midlatitude ionosphere dynamics from the ionosonde chain during strong SSW events, the results differ according to relative location of the stations with regard to the stratospheric zone.

The ionosphere is heavily affected by the geomagnetic activity, which is reflected in the electron concentration profile changes, most visibly in the F region. Both ionospheric density increase as well as decrease can occur during a magnetic storm in middle latitudes; they are called ionospheric positive and negative storms, respectively. The negative ionospheric storms are caused by a decrease in atomic oxygen density leading to a decrease in oxygen ion concentration and increase in the molecular nitrogen density leading to an increase in the loss rate. Both density changes (decrease in atomic oxygen density and increase in molecular nitrogen density) thus contribute to the resulting decrease in the ionization density in the F region. Positive ionospheric storms are typically explained in terms of traveling atmospheric disturbances with equatorward directed winds.

Traveling atmospheric disturbances (TADs) may also lead to enhancement in electron concentration in the F2 region [39,40]. The auroral thermosphere is heated due to Joule heating by ionospheric currents and enhancement of auroral precipitation. Pressure gradient from the pole toward equator changes the wind pattern and contributes to horizontal transfer toward middle and low latitudes. Atomic oxygen as a lighter gas is more mobile compared to heavier molecular nitrogen and therefore is transported farther from the auroral zone toward the equator.

The charged particles movement constrained to magnetic field leads to increase in elevation of the ionosphere maximum of ionization and consequently into increase in the maximum electron density as the gases responsible for the ionization loss (molecular nitrogen and oxygen) have much lower scale height than atomic oxygen which contributes to the charged particles production. The most negative disturbance is thus observed directly in the heating zone whereas the positive effect can be observed in lower latitudes. The storm-induced electric field can play important role in the ionospheric response [41].

The effects of electric fields in formation of the positive phase of ionospheric storms is discussed in [42]. The prompt penetration electric field is driven by the solar-wind-magnetospheric convection. The disturbance dynamo field is driven by the thermospheric wind including both a relatively fast component (2–3 h after SSC) and a relatively slow one (3–12 h after SSC). Both mechanisms lead to plasma $\mathbf{E} \times \mathbf{B}$ drifts and F2-layer uplifting. The interaction of two mechanisms is rather complicated [43,44].

Both described mechanisms (negative and positive storm) can serve as basic descriptions of the observed electron density changes during geomagnetic storms; however, many further issues can complicate the effect on the ionosphere [39].

## 2. Data and Methods

### 2.1. SSW Parameters and Geomagnetic Situation

The most important parameters describing SSWs are the zonal mean zonal wind at $60°$ N and polar temperature at 10 hPa. According to these two parameters, we can decide what type of SSW occurs in the stratosphere [45]. Main characteristics of the analyzed SSWs in January 2009, February 2018, and December 2018 to January 2019 are shown in Table 1. Their evolution is shown in Figure 1. Information in Table 1 and Figure 1 is derived

from the ERA5 (European Center for Medium-Range Weather Forecast Reanalysis 5 [46]) reanalysis data.

**Table 1.** Characteristics of the SSW events. $T_{max}$ stands for maximum polar temperature.

| Start Date | $T_{max}$ (K) | Date of $T_{max}$ | End date | Type |
|---|---|---|---|---|
| 19 January 2009 | 270 | 23 January 2009 | Early March 2009 | Split |
| 9 February 2018 | 244 | 17 February 2018 | 3 March 2018 | Split |
| 18 December 2018 | 265 | 28 December 2018 | 29 January 2019 | Split |

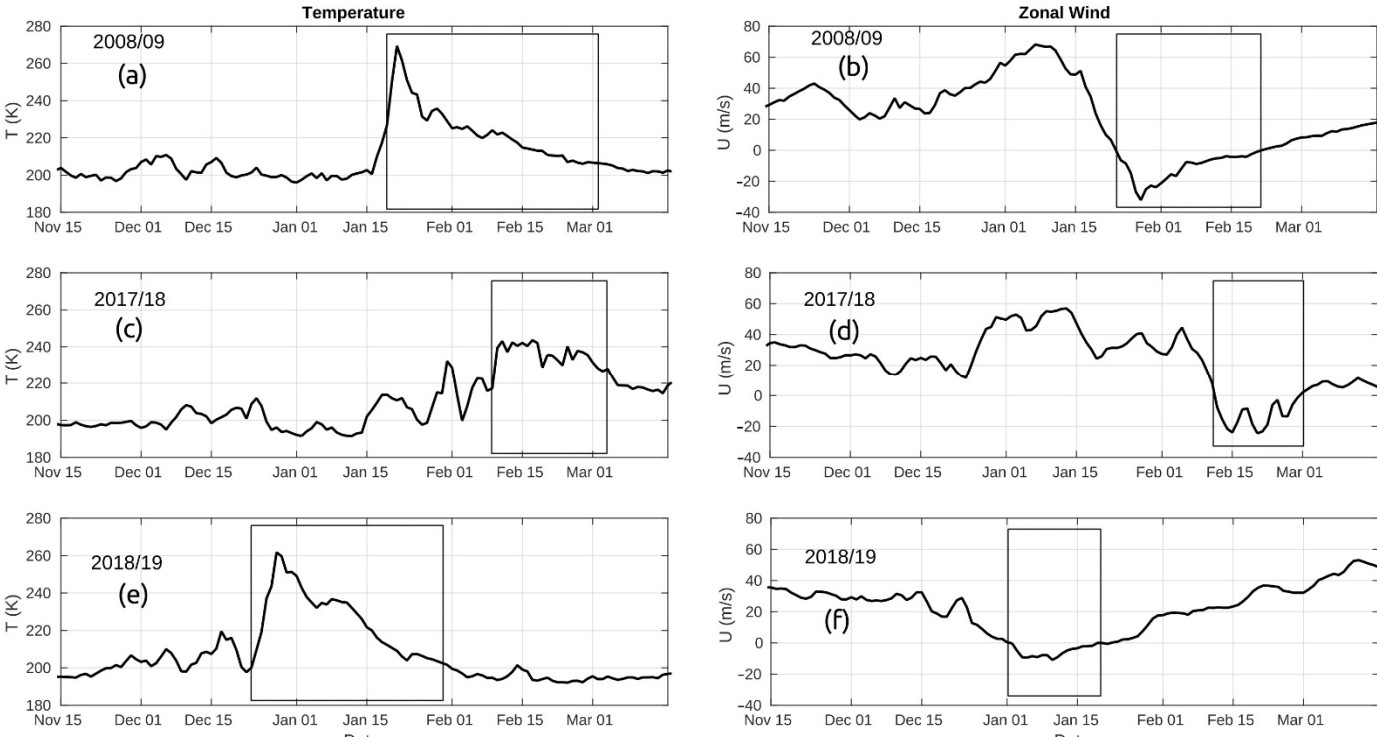

**Figure 1.** Polar temperatures at 10 hPa (left panels (**a**,**c**,**e**)) and zonal mean zonal winds at 60° N (right panels (**b**,**d**,**f**)) for winters 2008/09, 2017/18, and 2018/19. The rectangles denote SSW periods according to the temperature and periods of zonal wind reversal.

Upper panels of Figure 1 show the polar temperature at 10 hPa (panel (a)) and zonal mean zonal wind (panel (b)) during the January 2009 SSW. The beginning of SSW occurred on 19 January when we observed a strong increase in polar temperature at 10 hPa (by about 60 K in a week). The maximum temperature occurred on 23 January when its value reached 270 K. A strong zonal wind reversal started on 24 January. The pronounced maximum of easterly wind occurred on 28 January. The end of SSW according to temperature criterium is difficult to determine. The SSW finished in late February to early March, while the zonal wind returned to westerly on 22 February.

The February 2018 SSW took place after a 4 year hiatus in major warmings after the January 2013 event [47,48]. Middle panels in Figure 1 show polar temperature (panel (c)) and zonal mean zonal wind (panel (d)). The SSW started approximately on 9 February 2018 with an increase in polar temperature of about 25 K in a week. The multipeak maximum in temperature occurred during the 12–18 February period when the highest temperature at 10 hPa reached 244 K on 17 February. The zonal wind reversal started on 11 February. The double-peak amplitude maxima of the reversed zonal wind were on 15 and 20 February. In terms of temperature response, this SSW ended approximately on 3 March 2018, whereas zonal wind returned to westerly on 1 March 2018. We can consider this SSW as a major

because the WMO definition [45] is fulfilled, and we can observe zonal wind reverse at 10 hPa. This major SSW is weaker than the 2009 SSW, but we have to notice that SSW in 2009 is the strongest observed SSW in history so far [49].

The 2018/2019 SSW was the most recent major warming event [50]. Bottom panels in Figure 1 show polar temperature at 10 hPa (panel €) and the zonal mean zonal wind at 60° N (panel (f)). The SSW started on 18 December with an increase in polar temperature (more than 60 K in a week). The maximum of the SSW occurred on 28 December when the temperature at 10 hPa reached 265 K, and the warming finished around 29 January. The zonal wind reversal took place on 2 January 2019. The maximum zonal wind reversal was on 10 January, and wind returned to westerlies around 20 January. The temperature increase is comparable to the major SSW of 2009, but the zonal wind reversal is much weaker and shorter than in 2009. All three SSW events were major SSWs of the split not displacement type.

Geomagnetic Kp index is shown in Figure 2. Generally, the geomagnetic activity is low to moderate as can be expected for periods of very low solar activity. For most of the time, the Kp index was below Kp = 3 for all three periods around the SSWs. However, increased geomagnetic activity was observed for limited periods of time (more in sections Results and Discussion).

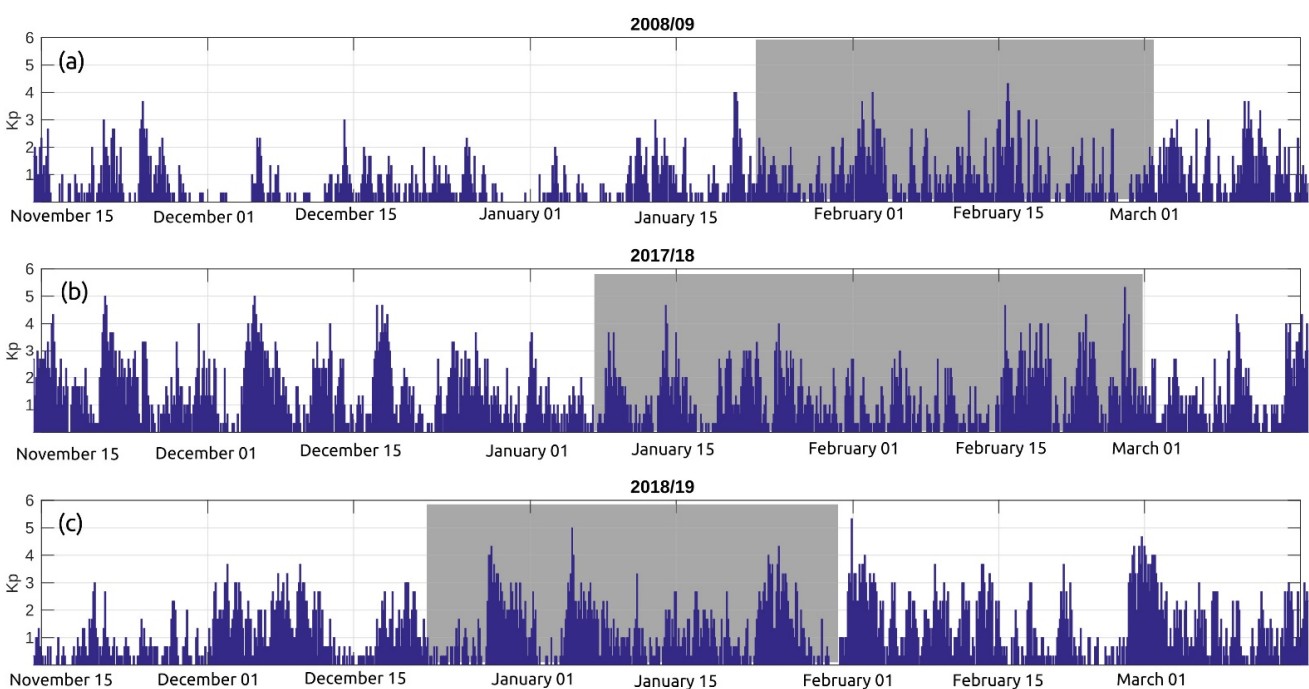

**Figure 2.** Kp indices corresponding to periods as in Figure 1. Time periods of SSWs according to stratospheric criterium are denoted as grey areas.

### 2.2. Digisonde Derived Parameters

The vertical sounding was performed using the Digisonde DPS-4D located in Julius-ruh, Germany (54.6° N, 13.4° E), Dourbes, Belgium (50.1° N, 4.6° E), Pruhonice, Czech Republic (50° N, 14.5° E), and Roquetes (Ebro), Spain (40.8° N, 0.5° E). The DPS-4D Digisonde allows for complex usage of the measurement. It consists of measurement using the ionograms, analysis of direction of the reflected signal, and methods based on Doppler shift measurement [51]. Using manual scaling of the ionograms, we obtained critical frequency of F2 layer (foF2), the peak height of F2 layer (hmF2), the critical frequency of E-layer (foE), the height of E-layer (hmE), and electron density profiles. Further, we analyzed the dynamics of the ionosphere using the directograms and the Digisonde Drift Measurement (DDM), e.g., [52].

The directogram allows for a more detailed analysis of the temporal evolution of nonvertical echoes distribution. The time is plotted on the *y*-axis. The *x*-axis shows the horizontal distance of the detected echoes from the vertical direction in east–west projection. Echoes detected in the west, northwest, and southwest directions are plotted in the left part of the directogram, and echoes from east, northeast, and southeast in the right part. The central line of the directogram corresponds to the vertical reflections and is usually deleted from the directogram to emphasize nonvertical signals. Under real conditions, the ionosphere is deformed to some extent which allows receiving of signals from areas distant from the zenith point. Display of nonvertical echoes helps in evaluation and interpretation of ionograms under spread F condition or other irregular ionograms [51], and in our case, the number of reflections is used as an overview of the wave activity in the F2 region [53]. The final result of Digisonde drift measurements (DDM) is the vector of drift velocity in the studied region above the station for a given measurement time. We studied the average velocity components during the studied intervals and compared them to the reference values for Pruhonice station [54].

### 2.3. Maps of TEC and Rate of TEC Index (ROTI)

Analysis of GNSS TEC dynamics is performed on the basis of global ionospheric maps (GIM) in IONEX format. The maps of global coverage provided by Center for Orbit Determination in Europe (CODE) and Technical University of Catalonia (UPC) are freely available at CDDIS server (ftp://cddis.gsfc.nasa.gov/gps/products/ionex/, accessed on 1 September 2020). The spatial resolution of the maps is $2.5° \times 5°$ in latitude and longitude, and temporal resolution is 2 h (1 h since 2015) for CODE and 15 min for UPC (UQRG). We also use the regional ionospheric maps provided by the Royal Observatory of Belgium calculated only for the European region (15° W–25° E 35–62° N) with the resolution 0.5° both in latitude and longitude for each 15 min. These maps are available at ftp://gnss.oma.be/gnss/products/IONEX/, (accessed on 1 September 2020) since 2012. To present dynamics of the ionosphere, we use global and regional maps to calculate an average TEC value over a given region and present time-versus-date scatter plots indicating average TEC by color.

Due to the temporal resolution, TEC maps do not present changes of TEC with periods below about 1 hour, and they do not provide information on traveling ionospheric disturbances (TIDs). The rate of TEC index (ROTI) allows analyzing the average intensity of such variations. Cherniak et al. (2018) presented a method for calculation of ROTI maps, and these maps are freely available at the CDDIS server together with TEC maps. The authors present the spatial ROTI behavior in dependence on magnetic local time (MLT) and corrected geomagnetic latitude (MLAT). Each ROTI map is generated for a specific day, within a 00–24 MLT temporal frame and a geomagnetic latitude range of 50°–90° N, with the corresponding cells of 2° both in magnetic latitude and longitude (equivalent to 0.13 h (8 min) MLT). Since a value in every cell is calculated by averaging all ROTI values covered by this cell area, the resulting ROTI value for every cell is proportional to the irregularity occurrence probability.

## 3. Results

### 3.1. SSW 2009 (19 January—Early March)

Ionospheric response in midlatitudes over Europe is represented here by a Digisonde measurement in Pruhonice, Czech Republic. Other European Digisondes used, Roquetes and Dourbes, show similar pattern to Pruhonice Digisonde Measurements. Further, we show TEC in the mid-European region (15° W–25° E, 35–62° N). Figure 3 shows the evolution of electron density profilograms at Pruhonice. Sequence of profilograms reveals a relatively stable course during the studied period with several clearly increased values (of about 1 MHz larger) of plasma frequency (the parameter directly connected to the electron density) around the peak height of F2 layer for 24 January 2009 as marked by an arrow. This increase in foF2 occurred one day after the detection of maximum temperature on 23

January and just on the day of the zonal wind reversal. We observed increased day-to-day changes in the foF2 parameters during the increased temperature period. Decrease in peak height of the F2 region was observed on 28 January, i.e., around the maximum speed of reversed zonal wind, as well as on 1 and 2 February (i.e., still during the reversed phase of zonal wind). In the E region, foE did not vary significantly; however, we observed a slight decrease in hmE by a few km for the day of maximum temperature as well as for the reversed zonal wind maximum. The geomagnetic activity for several days before and around this maximum was very low (Kp = 0–1) as shown in Figure 2, which means that the observed effects of SSW were without a geomagnetic "contamination".

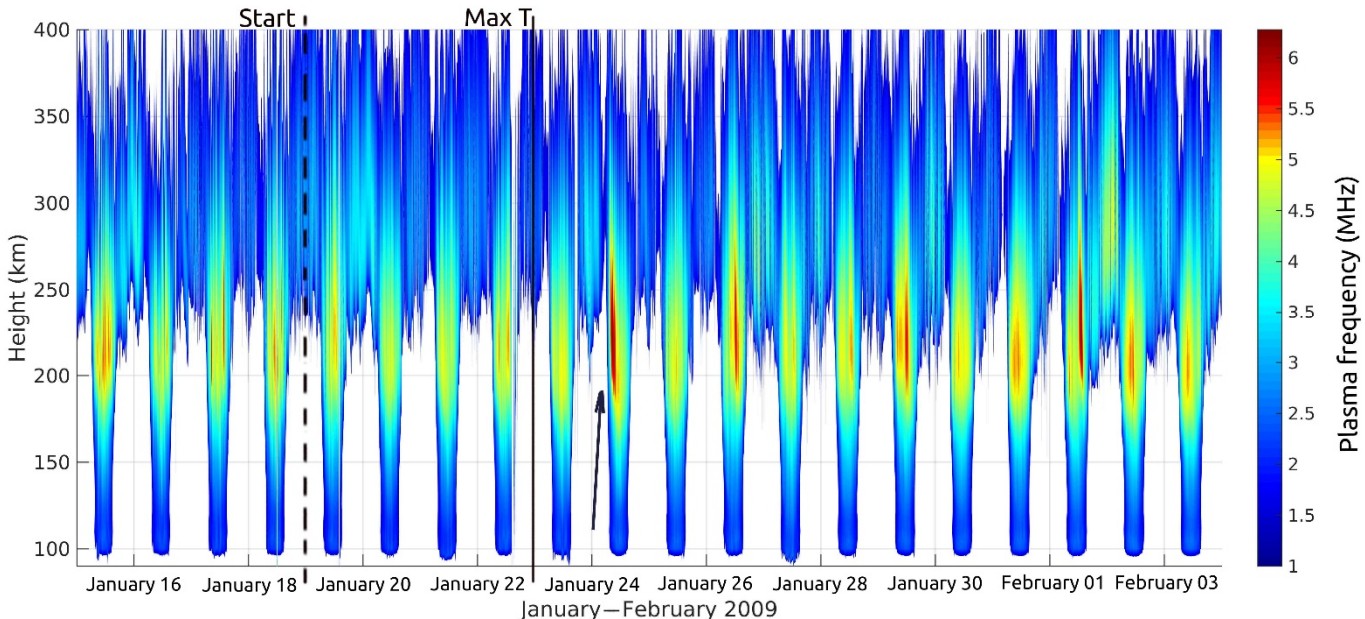

**Figure 3.** Plasma frequency profiles at Pruhonice from 15 January to 3 February 2009. The dashed and solid line denote the start of SSW and maximum in stratospheric temperature, respectively. The arrow denotes an increase in plasma frequency on 24 January.

The described enhancement in electron density on 24 January during geomagnetically quiet time was followed by a slight increase in Kp on 26 January (Kp $\leq$ 3+). It resulted in the increase in hmF2 on 26 January and night 26/27 as well as increase in electron density around hmF2 observed using the Pruhonice Digisonde (Figure 3). Figure 4 shows critical frequencies foF2 at four European stations (from top to bottom: Juliusruh, Dourbes, Pruhonice, and Roquetes). The actual course is compared to median hourly values from a week of preceding geomagnetically quiet period. The highest increase in foF2 is observed at Roquetes, and less pronounced effect is seen at Pruhonice and Dourbes stations. We observed slight foF2 decrease over Juliusruh in the initial part of the SSW (18–21 January) and practically no ionospheric change on 24 January. Therefore, the magnitude of the increase indicates the latitudinal dependence. It should be also mentioned that during the SSW 2009, the ionospheric F1 layer has often been observed, which is a rather rare case [55,56]. Typically, during winter, the F layer does not split into F2 and F1 regions.

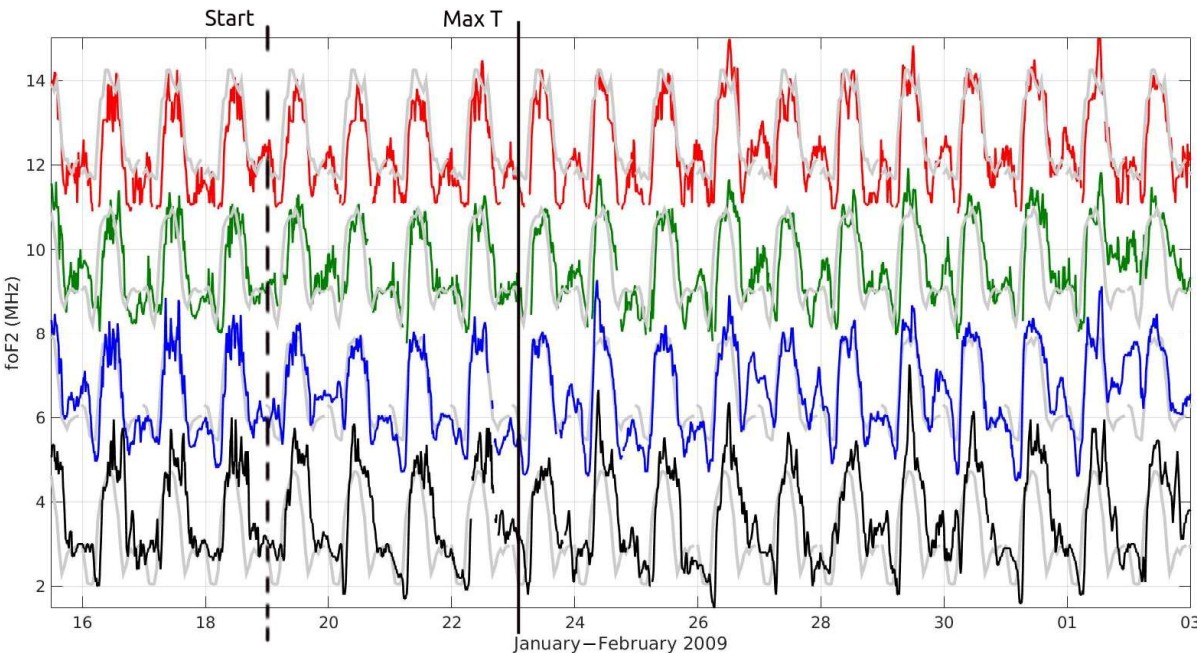

**Figure 4.** Critical frequencies at Juliusruh, Dourbes, Pruhonice, and Roquetes (red, green, blue, and black, from top to bottom). Light gray lines denote quiet course of foF2 for individual stations. Dashed and solid vertical lines denote onset of SSW and maximum in stratospheric temperature, respectively. Increase in daily foF2 values is significant at three stations on 24 January. No change in foF2 at Juliusruh was observed.

Between 21 and 22 January, we observed a very slight decrease in TEC compared to preceding values, most significantly during morning hours (around 00-07 UT). No significant changes in average TEC over Europe on 24 January or connected to polar temperature or zonal wind changes or extremes were observed (Figure 5). On the other hand, an enhancement of TEC occurred on 26 January, day of weak geomagnetic storm (in agreement with daily foF2 increase at all four ionospheric stations).

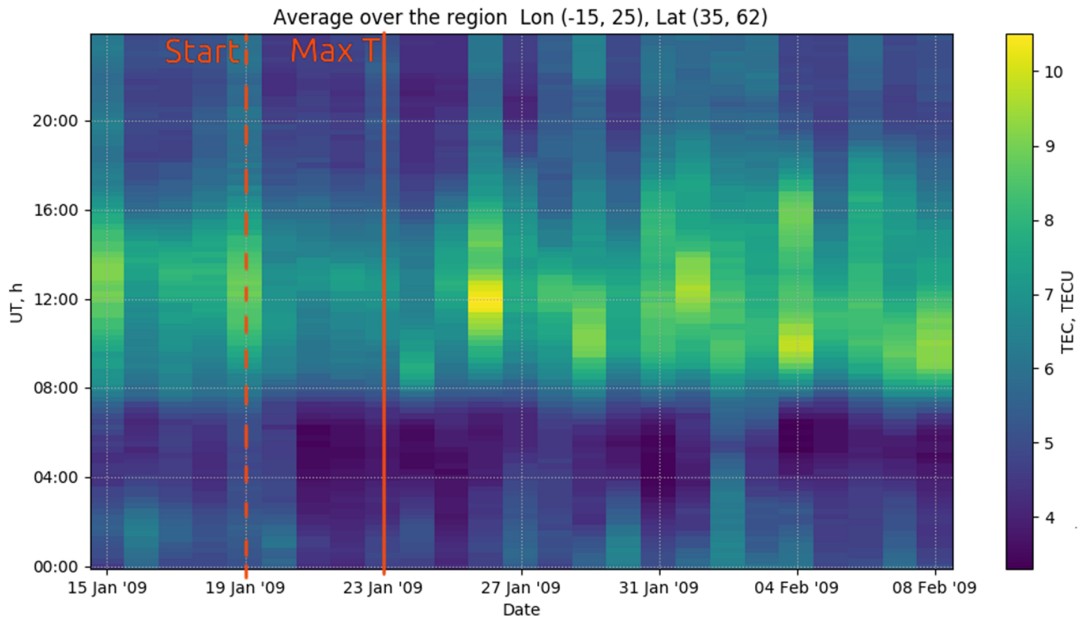

**Figure 5.** Average vertical TEC over midlatitude Europe (35–62° N, 15° W–25° E) by UPC data. Vertical axis denotes time in UT; colorbar is in TECu. Dashed and solid lines denote onset of SSW and maximum in stratospheric temperature, respectively. No significant increase in TEC on 24 January was observed. In Central Europe LT = UT + 1.

The directogram from Pruhonice station (Figure 6) reflects the ionospheric activity in the F2 region. The most pronounced activity is visible on 21 January, 23 January (maximum of stratospheric temperature), 24 January (beginning of zonal wind reversal), and 1–2 February (not connected to the maximum in zonal wind reversal but in the period of the reversed/easterly direction).

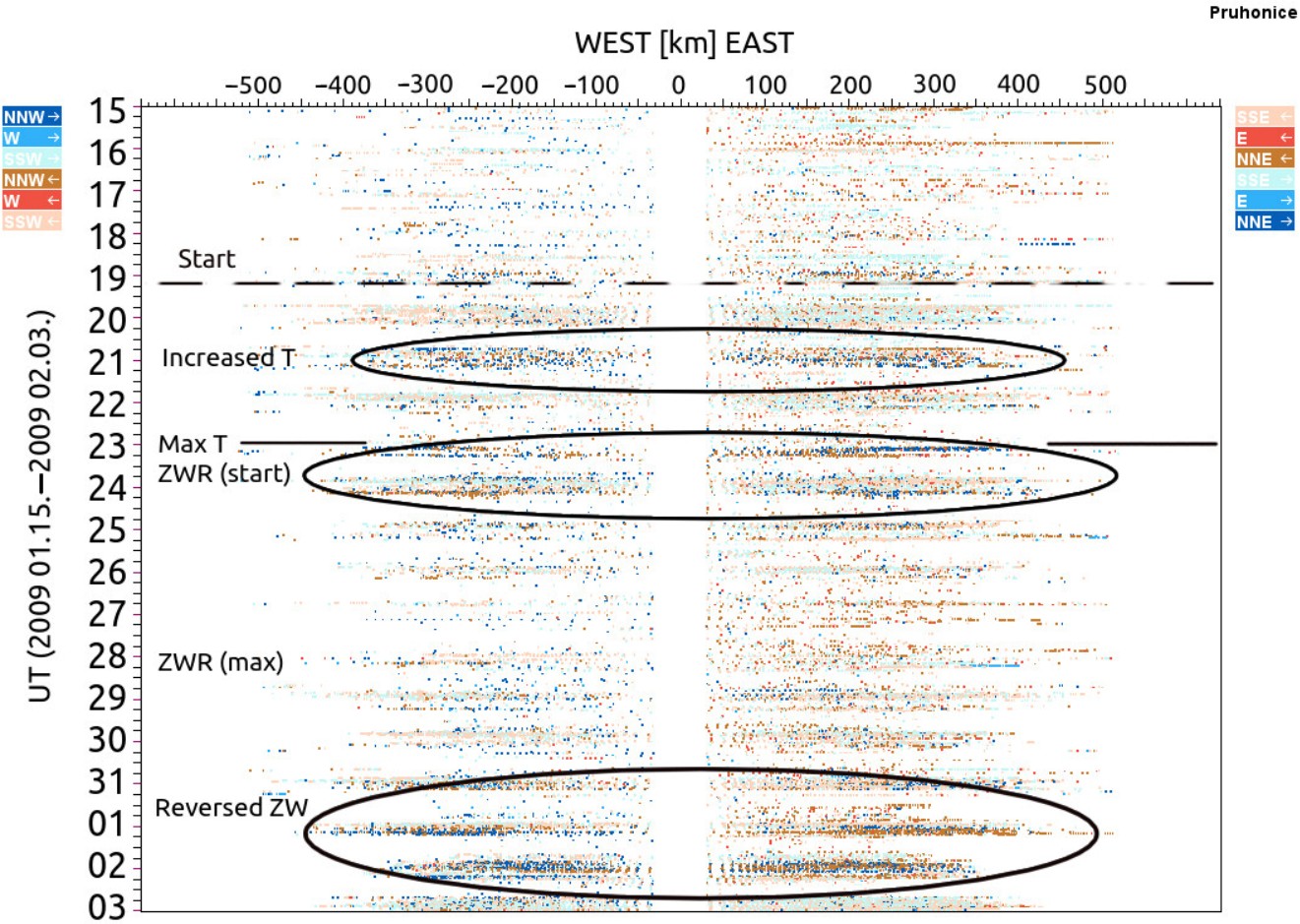

**Figure 6.** Directogram from the Pruhonice station. The color intensity denotes amplitude in the received signal; the color shows direction of the movement of the ionospheric plasma. Start and Max T denote onset of SSW and maximum in stratospheric temperature, respectively. ZWR (start), ZWR (max) and reversed ZW denote start of zonal wind reversal, maximum speed during zonal wind reversal period, and period of zonal wind reversal before returning to normal situation, respectively.

### 3.2. SSW 2018 (9 February—3 March)

Observations of foF2 at the four ionospheric stations reveal a sharp increase in daily maximum of foF2 on 17 February 2018 (Figure 7). The difference between observed values and median values increases with decreasing geographic latitude, i.e., similarly to the 2009 SSW, the highest response is seen at Roquetes station and the smallest but still noticeable at Juliusruh. In addition, 17 February was the day with the maximum temperature of this SSW (the whole interval of significantly enhanced temperature is between 12 and 18 February) as well as it occurs between the two maxima of reversed wind on 15 and 20 February. The ionospheric response to the SSW is however complicated by an increase in the geomagnetic activity. The geomagnetic activity on 17 February reached Kp 4− (Figure 2) in the morning before the electron density enhancement. Geomagnetic activity on 19 January 2018 (before SSW) was comparable (up to Kp = 4) without any significant effect in foF2. On 23 January 2018, again during increased geomagnetic activity, a rather small change in foF2 was

observed at all four stations (Kp reached 4+) compared to the 17 February. This suggests that the enhancement of foF2 on 17 February cannot be explained solely by the observed geomagnetic activity, and that the role of SSW is substantial, probably principal. The observed electron density (foF2) enhancement on 17 February 2018 is the main ionospheric feature of this SSW event. The peak height hmF2 on 17 February does not show significant difference from preceding days. The directogram (Figure A1 in Appendix A) does not show clear connection of ionospheric wave activity to particular stages of the SSW.

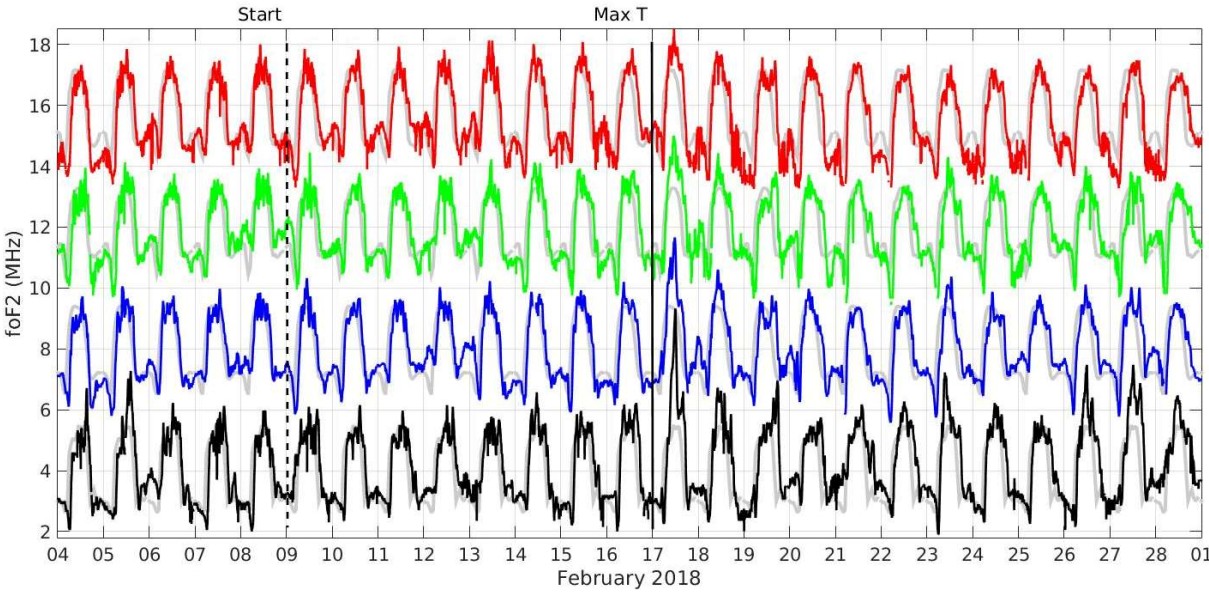

**Figure 7.** Critical frequencies at Juliusruh, Dourbes, Pruhonice, and Roquetes (red, green, blue, and black, from top to bottom). Light gray lines denote median values for individual stations. The dashed and solid vertical lines denote the starts of SSW and maximum in stratospheric temperature, respectively. The most dominant feature is a short-time increase in maximum daytime values on 17 and 18 February at all stations with higher effect at three southern located stations and less pronounced response in Juliusruh.

The observed as well as simulated average TEC over European region and the ROTI index development are shown in Figure 8. The maximum observed TEC over the whole interval was observed on 17 February. The average TEC maximum (panel a) coincides with foF2 data as well as with the TEC obtained from the Pruhonice ionosonde. No other comparable enhancement of TEC was observed during the period 4 February–1 March.

The evolution of TEC over European region was simulated with WACCM-X (the whole atmosphere climate community model—eXtended [57]). The panel (b) in Figure 8 shows that the maximum of simulated TEC occurred around 17 February, on the same day as the observed maximum of observed TEC and foF2. The comparison of the model with observed TEC is used for evaluation and validation of WACCM-X. The modeled TEC may be compared only in a qualitative sense with the observed TEC because the upper boundary of WACCM-X (~500–800 km) is significantly lower than the altitude of GPS satellites (~20,000 km), which contributes to substantially lower values of TEC in the model as compared to observations. The model simulations overall do support the observations in a qualitative sense, which suggests that WACCM-X can capture most of the day-to-day variability in the thermosphere-ionosphere system. More details about simulations of ionospheric effects of SSWs of 2017/2018 and 2018/2019 by WACCM-X may be found in [58].

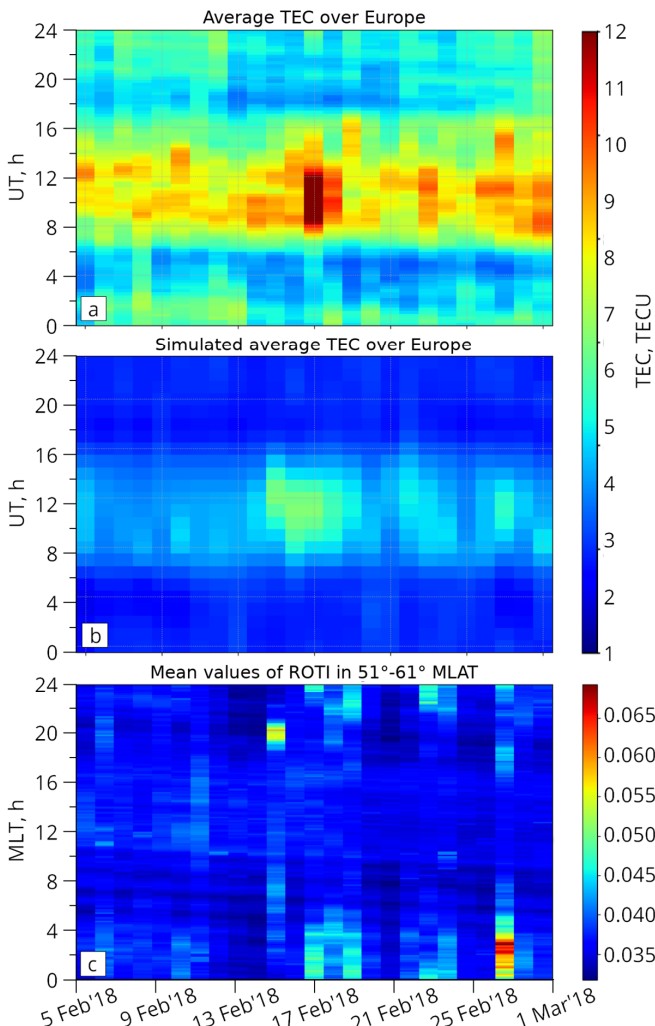

**Figure 8.** Panel (**a**): the observed average TEC values over the European region for SSW 2018; panel (**b**): the modeled TEC values for the European region; and panel (**c**): ROTI variations derived from ROTI maps at geomagnetic latitudes 51–61° N.

Using daily maps, we calculated average ROTI over all magnetic latitudes (MLAT) to present day-to-day variations during a given period. The average ROTI values $R_{av}$ for all MLTs are shown to be somewhat enhanced on February 15, 17, 23, and mainly 27 (panel (c) in Figure 8) essentially at night, whereas TEC enhancements occurred near a local noon. Periods of the increased Rav are in a good correspondence with increased Kp values (Figure 2) and are likely driven predominantly by geomagnetic activity without a clear connection to the SSW. Thus contrary to TEC and foF2, ROTI seems to be relatively insensitive to the 2018 SSW.

We observed the so-called spread conditions in the F region between 17 and 21 February 2018, i.e., during and just after the days with observation of maximum temperature at the level 10 hPa (an example of such an ionogram is in panel (a) of Figure 9). The spread F condition is a result of nonplanar ionosphere when the transmitted signal from the ionosonde is reflected from undulated isodensity surface leading to diffused shape of the registered echo. The ionograms before 17, and after 21 February are without presence of spread condition (the later example is in panel (b) of Figure 9). The DDM results do not show any noticeable deviation from expected values described in [52] (not shown here). The directograms show a slight increase in wave activity in AGW domain for the time range 17–22 February 2018, i.e., during and a few days after the peak in stratospheric

temperature, see Figure A1 in Appendix A. We did not observe significant changes in the peak height or critical frequency of the E region.

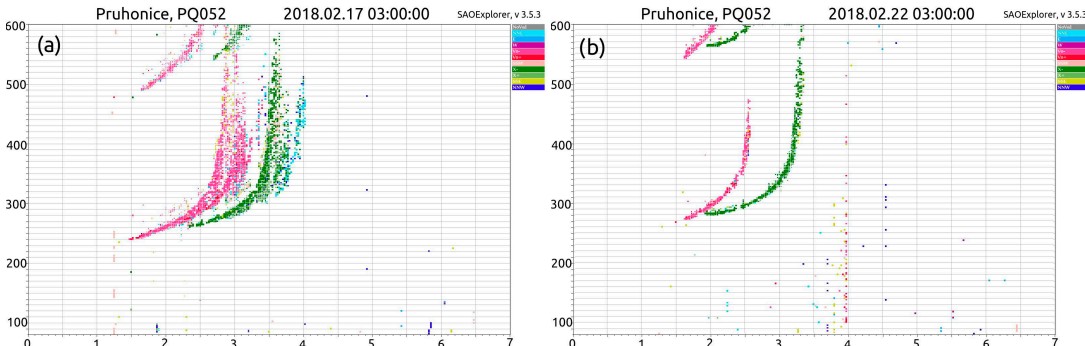

**Figure 9.** Panel (**a**): spread F conditions on 17 February 2018. Such types of ionograms are results of wave-like activity in the ionosphere causing departures from normal/horizontal stratification. Red and green colors denote ordinary and extraordinary reflections, respectively. Panel (**b**): ionogram from the corresponding time on 22 February 2018 without any spread conditions.

### 3.3. SSW 2018/2019 (18 December 2018–29 January 2019)

Similarly to the February 2018 SSW, we observed significant increase in foF2 (Figure 10) and TEC (Figure 11) on 28 December 2018, which was a day of the stratospheric temperature peak; it was accompanied by a slight increase in hmF2. As in the case of the February 2018 SSW, the situation was overlaid with increased geomagnetic activity as the day 28 December was influenced by geomagnetic active conditions (Kp ≤ 4+). For the time interval between the day before 28 December and two days after we observed significantly increased noon values as well as an increase in evening hours compared to median hourly values from a week computed from quiet time prior the SSW event. The ionospheric data from the week preceding the SSW maximum, i.e., before 25 December 2018, and from the periods one and two weeks after, i.e., after 31 December 2018, do not depart from median values. Moderate geomagnetic storm (Kp = 5) occurred on January 5 (Figure 2). It resulted in a weaker effect on foF2 values than observed on 28 December. Hence, we suggest that the effect of SSW on 28 December on foF2 and TEC enhancement is important and cannot be overlooked.

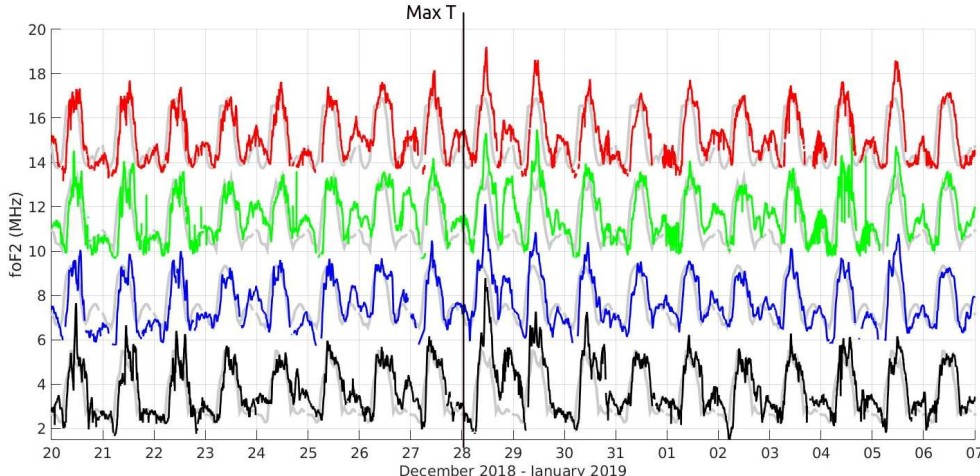

**Figure 10.** Critical frequencies foF2 at Juliusruh, Dourbes, Pruhonice, and Roquetes (red, green, blue, and black, from top to bottom). Light-gray lines denote median values for individual stations. The most dominant feature is a short-time increase in maximum daytime values around 28 December (day of temperature maximum, denoted as solid vertical line). The beginning and end of SSW are outside of the shown interval.

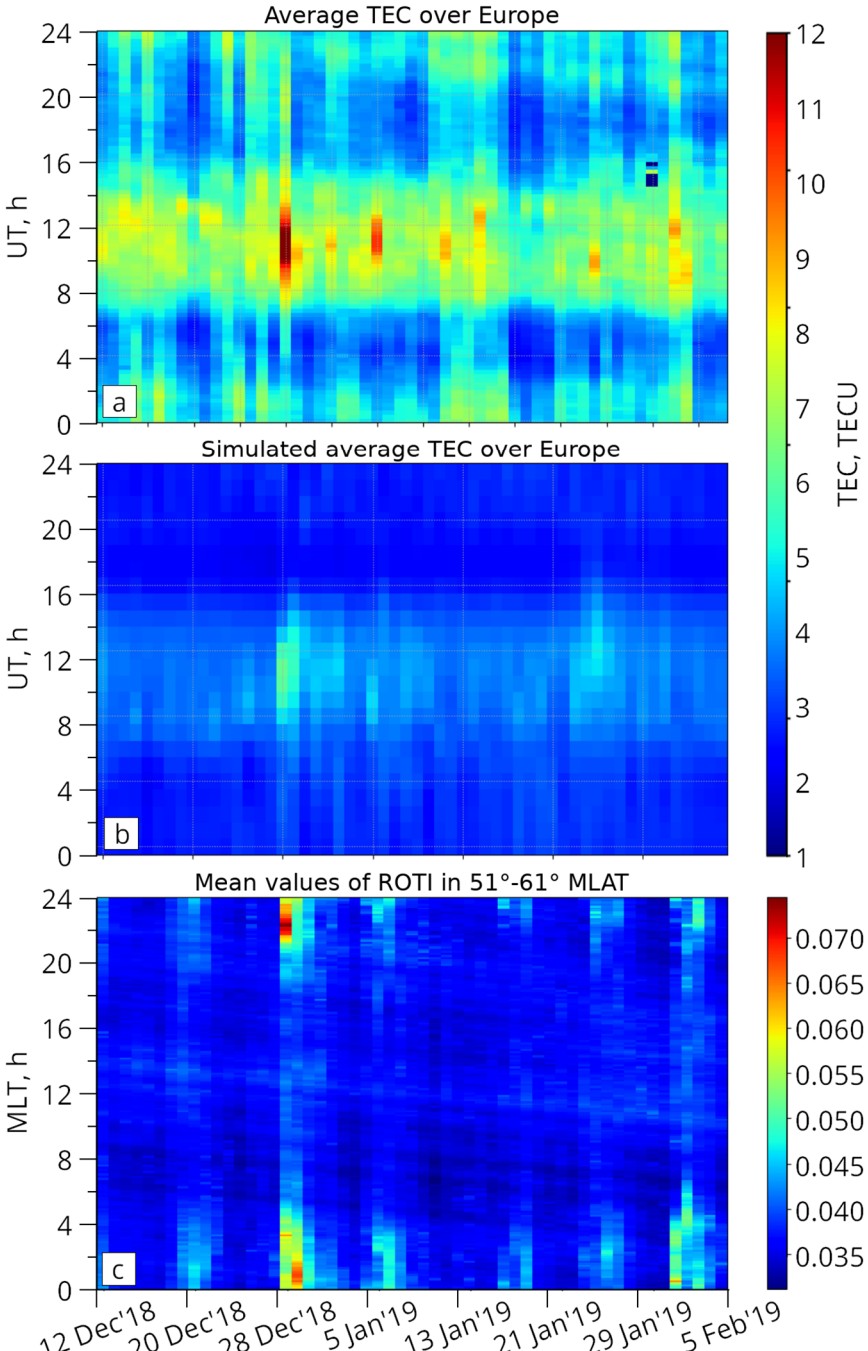

**Figure 11.** Panel (**a**) average TEC over Europe for SSW 2018/19, (**b**) modeled TEC values over Europe, and (**c**) average ROTI within 51–61° N.

Panel (a) in Figure 11 shows the average TEC over Europe. Increase in TEC for several hours on 28 December 2018 is observed. Much weaker increase in TEC on 5 January is probably an effect of a geomagnetic storm (Kp = 5). The observed evolution of TEC is qualitatively in agreement with the modeled evolution of TEC (Figure 11, panel (b)). The enhancement of TEC on 28 December 2018 near noon (11 UT) is accompanied by the increase in average ROTI values from the geomagnetic latitudes between 51° N and 61° N at night (Figure 11, panel (c)). A slight increase in ROTI on 5 January is related to the moderate geomagnetic storm of this day. The wave activity observed using the directograms (see Appendix A, Figure A2) is increased in the whole interval between 12 and 28 December 2018 except for 23 and 24 December with a significant maximum on

28 December. Except for the day 28 December, influenced by the geomagnetic activity, the SSW peaks in temperature and zonal wind velocity are not clearly reflected in the directogram results. The DDM results for vertical component of drift velocities do not differ significantly from reference average values described in [52].

Figure 12 shows comparison of the TEC maps from 28 December 2018 (the day of maximum stratospheric temperature, upper left panel) with days of similarly increased Kp on 5 January, 23 January, and 24 January. The TEC distribution in middle latitudes on 28 December was significantly different from the other days, whereas in high latitudes, it looks similar. This observation is in good agreement with comparison of TEC maps from early morning hours (between 00 and 06 UT) on 28 December 2018 and on 5 January 2019 when the TEC values over Europe reached significantly higher values for the 28 December 2018 under conditions of practically identical geomagnetic activity (see Figures A3 and A4 in Appendix B). This seems to account for the finding that the high-latitude ionospheric effect on 28 December 2018 is essentially of geomagnetic origin, whereas the midlatitude effect is driven by the SSW.

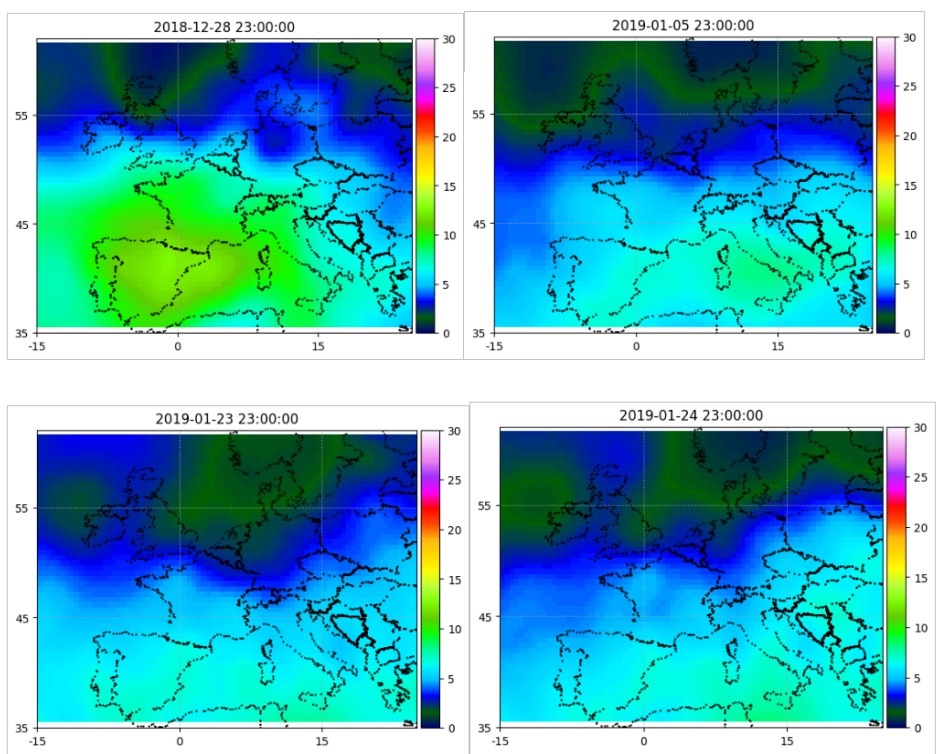

**Figure 12.** Comparison of TEC maps from the day of the peak of foF2 and TEC (28 December 2018, Kp 3+) and three days with similar geomagnetic activity in January 2019 (5 January, Kp 5, 23 January, Kp 4, 24 January, and Kp 4+, respectively).

## 4. Discussion

During the 2009 SSW, just one day after the maximum in temperature on 23 January 2009, significant enhancement in critical frequency foF2 was registered for three stations Dourbes, Pruhonice, and Roquetes on 24 January, which corresponds to a day of onset of zonal wind reversal. The magnitude of foF2 changes indicates latitudinal dependence. Almost no increase is detected within foF2 course monitored in Juliusruh, the northernmost station involved in the analysis. This might suggest that the effects of SSW are observed up to at least 50° N in latitude, whereas the ionospheric response to SSW at higher latitudes is much weaker and for Juliusruh located at 54° N negligible. Clearly, the effect of the 2009 SSW is amplified toward southward located stations. We did not observe significant change in hmF2 for SSWf maximum temperature or zonal wind reversal maximum. The

ionospheric wave activity deduced from the directogram in Pruhonice is pronounced mainly during the period of increased temperature and onset of zonal wind reversal. It does not show signs of increased activity for the zonal wind reversal maximum, whereas a short period of increased wave activity is observed in the period of decreasing reversed zonal wind speed. As the geomagnetic activity on 24 January and a few days before was extremely calm, the enhanced electron density on 24 January cannot be attributed to geomagnetic activity. An increase in hmF2 during minor geomagnetic storm of 26 January and night of 26/27 January 2009 as well as increase in electron density around hmF2 can be explained by the effects of geomagnetic storm on the neutral atmosphere affecting the ionosphere, namely due to thermospheric winds and traveling atmospheric disturbances (TADs) [39,59]. We observed a slight decrease in hmE corresponding to the 2009 SSW temperature maximum as well as zonal wind reversal maximum compared to background values (Figure A5 in Appendix C). However, we cannot clearly state whether these hmE changes are connected to the SSW effects. The parameter foE did not vary significantly for most of the studied period.

Remarkable identified feature is a growth in plasma profile on 24 January (Figure 3), foF2 critical frequency data (Figure 4) as well as directogram results (Figure 6), compared to practically no change in average TEC (Figure 5) for the same day. Significant differences in evolution of foF2 and TEC were sometimes observed and reported, particularly during geomagnetic storms (e.g., [60]). The difference between maximum plasma concentration and integral value of electron concentration may be explained by redistribution of electrons along the electron density profile and/or field lines leading to increase in foF2 but keeping TEC unchanged. As we mentioned in the Results, the split of F to F1 and F2 is rather unusual for the winter ionosphere. The formation of the F1 layer is associated with the temperature regime of the lower thermosphere leading to changes in ratio between atomic oxygen and molecular gas. These variations cause a change in the rate of recombination processes, which can alter concentrations of the main components of the thermosphere [61]. This can be explained as the effects of wave activity enhancement. Our observation is in agreement with the results of [61] who reported increase in occurrence of F1 during SSW events and indicate that the wave activity enhancement in the underlying atmosphere can contribute to the occurrence of the midlatitude F1 layer.

The 2009 SSW event occurred under very quiet geomagnetic and solar activity conditions, and therefore, the observed effects can be attributed to the SSW influence on the F2 region in middle latitudes; this general conclusion is consistent with earlier results (e.g., [29]). There are corresponding features in TEC observation for the SSW 2009 event in our observation and other works, namely decrease in TEC in the beginning of the SSW after 19 January 2009 up to the peak in stratospheric temperature [30]. Our observations of increase in foF2 around and after SSW temperature peak for SSW 2009 differ from results obtained by [25,36] who reported systematic decrease in foF2 after 19 January. The difference might be attributed to finding that Siberian and central European sectors were under different stratospheric situations. Using [62] (Figure 8, therein), we may see that the European sector lies in the area of higher geopotential compared to the sector corresponding to locations of stations described in [35,36] as well as Jicamarca or San Jose dos Campos reported in [37].

The interpretation of the February 2018 SSW ionospheric impact was rather complicated due to increased geomagnetic activity, and the question that remains to be answered is to what extent is the ionosphere influenced by the SSW. Corresponding behavior—increase in foF2 parameters on the four stations in our study—is identified similarly to the previous case 2009 SSW. The sharp increase in plasma frequency was detected by means of foF2 at Roquetes, Dourbes, and Pruhonice station as well as in TEC around noon hours for the day of 17 February 2018 (Figures 7 and 8). Noticeably, the foF2 increase on 17 February in Juliusruh was observed, but the increase was weaker than at the other three stations. However, this day is both a day of increased geomagnetic activity and also maximum stratospheric temperature, and it is located between the reversed zonal wind reversal

maxima (15 and 20 February). With no doubts, the ionosphere is under the influence of both the geomagnetic storm and SSW. To untangle these effects is not an easy task.

As mentioned in the Introduction, there are two possible scenarios for electron density during a geomagnetic storm—positive and negative. In the case of negative geomagnetic storm (decrease in electron concentration), the resulting observed electron concentration is determined by a decrease in electron concentration caused by geomagnetic forcing and an increase in electron density caused by SSW. Hence, one may conclude that the contribution of SSW is dominant if an increase in electron concentration is observed. In the case of a positive storm (increase in electron concentration), both geomagnetic and SSW contribute to the resulting observed growth of electron concentration.

There are several indirect clues that the SSW effects may play an important role for this period. First, similarly increased geomagnetic activity on 17, 19, and 23 February 2018 (Kp 4−, Kp 4, and Kp 4+, respectively) resulted in much weaker responses of foF2 (Figure 7) on 19 and 23 February for all stations compared to 17 February 2018. Using the TEC data, the largest change in average values of TEC for the European region was observed on 17 February and the other two days (19 and 23 February) show much weaker response. Differences in the ionospheric response could be explained as a combination of both geomagnetic and SSW forcing of the F2 region on 17 February, whereas on 23 February only the geomagnetic forcing was present. This again supports the idea of the important impact of SSWs on the ionospheric F2 region.

Additional argument supporting this idea is that the spread-F occurred in ionograms in Pruhonice station only in a limited time interval between 17 and 21 February 2018 coinciding with the overlap of maximum temperature and days around zonal wind reversal maxima. Contrary to that, during conditions of similarly increased geomagnetic activity after 21 February (e.g., on 23 February), the ionograms were without such strong spread-F phenomenon. We assume that the ionospheric wave activity in the F2 region as deduced from the spread condition ionograms was stronger on days with increased geomagnetic activity, when the SSW parameters reached their maxima, than during similarly increased geomagnetic activity outside of this interval. The ROTI values agree well with the geomagnetic activity and show maximum for the day of 19 February, and therefore, this maximum does not correspond to the foF2 daily peak. It seems that the average TEC is in this case apparently much more sensitive to a geomagnetically active day 17 February than to other geomagnetically active days (Figure 8), and the TEC increase is in a good correspondence with the SSW maximum phase. It can be deduced that the SSW played an important role in the observed ionospheric enhancement on 17 February 2018.

The most dominant ionospheric effect of the 2018/2019 SSW as deduced from the plasma frequency profiles and ionospheric parameters at all studied stations was a gradual increase for several consequent days followed by a decrease again in daytime electron density around hmF2 between 27 and 30 December with a significant peak on 28 December. The 28 December 2018 was a day of the maximum stratospheric temperature and the day of increased geomagnetic activity. The ionospheric response in foF2 and TEC on 28 December can be therefore explained as the result of combination of geomagnetic forcing and SSW forcing, but comparison with other days of enhanced geomagnetic activity in January 2019 suggests the dominant role of SSW at European middle latitudes contrary to high latitudes dominated by geomagnetic activity. The cross-correlation multiscale analysis [63] shows statistical negative correlation between Kp index and foF2 for six European stations (i.e., increase in Kp statistically leads to lower foF2). The analyses demonstrated that within the studied data from midlatitudes the scenario of negative storm is more probable to be observed. Both positive and negative deviations of foF2 have been observed under extremely low solar activity conditions of 2007–2009 independent on season and location [59]. The authors reported that positive effects on foF2 prevailed and were more significant. Hence, both scenarios of negative and positive storms should be considered.

During the 2018/2019 SSW the foF2 increase related to the peak of SSW started one day before the increase in geomagnetic activity. Compared to 28 December, ionospheric enhancement on 5 January (Kp = 5) shows noticeably smaller change in foF2 and TEC. Similarly, the increased geomagnetic activity on 24 January (Kp = 4+) does not show changes in the electron concentration. The average ROTI values correspond well with the geomagnetic activity deduced from Kp index. Except for 28 December, none of the mentioned intervals of increased ROTI values are related to significant change in TEC. As the ROTI values are connected to short-time changes in the ionosphere, it suggests that the TEC increase on 28 December 2018 might have been influenced or connected with the 2018/2019 SSW temperature peak. The rate of TEC index (ROTI) is calculated by averaging ROT values over a period of 5 min. That means that relatively long-term processes (hours) do not give a significant increment of ROTI, although it can be seen in TEC change. Moreover, the presented values are taken from global ROTI maps, and they were calculated by averaging values from areas of the same magnetic latitude. Therefore, high values in ROTI maps mainly represent frequent changes in TEC (minutes) observed simultaneously over the world.

The one suggested explanation is that the observed increase in both TEC and ROTI is produced by a combination of SSW and geomagnetic influence.

During all the three events, we detected an increase in wave activity by means of the directogram; however, only the 2009 SSW shows direct link between the directogram deduced activity and stratospheric temperature and zonal wind speed parameters. It is the only SSW event observed under quiet geomagnetic conditions. The state of the ionosphere during two recent SSW cases is influenced by minor-to-moderate geomagnetic storms. Hence, any of the observed effects may be caused by a combination of both geomagnetic and lower atmospheric forcing. The connection of directogram results and SSW parameters for the two recent events is not fully decisive; nevertheless, an increase in wave-like activity is evident. Directograms provide qualitative indication of the ionospheric behavior. Digisonde detects strong and off-vertical echo when the ionosphere is not horizontally stratified which means the isodensity planes depart from horizontal. Registered ionograms are often Spread-F type. Such a situation is often connected with propagating atmospheric waves in particular AGWs. (e.g., [64–66]). Resulting directograms clearly indicate increasing wave activity responsible for ionospheric irregularities and consequent off-vertical echo. Changes in color on directograms show fast shears in plasma motion within a rather short time as reported in [54,67].

The DDM results in all three SSW events show no clear link between plasma velocity and SSW parameters. The average velocity components from F2 region computed for different phases of the SSW do not show significant deviation from the expected undisturbed values for a given season. This finding has not yet been understood, but we assume that the plasma drift is forced on shorter scales (minutes to hours) connected to geomagnetic activity, whereas the SSW effects proceed on longer time scales of hours to days.

During both SSW 2018 and SSW 2018/2019, ROTI showed nighttime enhancement that can be attributed to geomagnetic disturbances. In addition, positive ionospheric storms are mainly a feature of winter seasons (see for example [41,68]. Both facts may indicate a dominant geomagnetic effect on the ionosphere. In the WACCM-X simulation results, shown here for the 2018 and 2019 SSWs, the geomagnetic forcing is included. In a companion study [18], we carried out two pairs of simulations for the 2018 SSW and SSW 2018/2019 in order to isolate the effects of geomagnetic and lower atmospheric forcing on the TEC variability. In the first simulation setup (S1), the TIE-GCM forced by WACCM-X is run in its default mode, and the obtained day-to-day ionospheric variability from this run includes the effects of both geomagnetic and lower atmospheric forcing. In the second simulation setup (S2), we turn off the geomagnetic forcing and carry out a similar run for both SSWs. For both events, the simulations show that the lower atmospheric forcing leads to an increase in TEC on days corresponding to the maximum stratospheric temperature.

## 5. Conclusions

We analyzed three major SSWs from years of deep minima of solar activity with the aim of estimating the role of the lower atmosphere forcing on the midlatitudinal ionosphere. SSW 2009 occurred during quiet geomagnetic conditions while the SSW events 2018 and 2018/2019 were affected by minor-to-moderate geomagnetic storms. Only limited number of papers has been studying ionospheric effects of SSWs at middle latitudes compared to low latitudes. Thus, novelty of our contribution is the analysis of 2018 and 2018/2019 SSW impacts on the midlatitude ionosphere over Europe and finding that SSW has a significant effect on the midlatitude ionosphere both under geomagnetically quiet as well as geomagnetically disturbed conditions. We observed significant ionospheric short-time changes for all three studied SSW events, and all of them exhibited an increase in the plasma frequency (foF2) on days of maxima of stratospheric temperature or very close to them, partly in coincidence with the reversed zonal wind that defines the occurrence of major SSW, as well. Significantly, foF2 increased by about 20–30% compared to reference days, and the increase was more expressed for stations at lower latitudes. Out of these three events, the 2009 SSW is the only case with the SSW forcing being the only source of the observed ionospheric changes. The February 2018 and December 2018/January 2019 SSWs represent situations when the ionosphere is jointly influenced by the geomagnetic and SSW forcing. The geomagnetic activity was in these two later SSWs indisputable. Our detailed analyses, together with observations in 2009 when the ionosphere was geomagnetically quiet, suggest that despite the geomagnetic activity the observed ionospheric effects are still significantly influenced by the SSW. Based on the analyses of three SSW events occurring during deep minima of solar cycles (2009, 2018, and 2018/2019), we conclude that the SSW considerably enhanced ionospheric disturbances in the middle-latitude European region.

**Author Contributions:** Writing-draft preparation, Z.M. and J.L.; methodology, Z.M. and I.E.; SSW description, M.K.; Digisonde measurement analysis, Z.M., D.K., and P.K.K.; GPS analysis, I.E.; TEC modeling, T.A.S. All authors have read and agreed to the published version of the manuscript.

**Funding:** This work was supported by the European Space Agency project VERA (VERtical coupling in Earth's Atmosphere at mid and high latitudes). IE work was performed with budgetary funding of Basic Research program II.16. T.A.S was supported by the Alexander von Humboldt Foundation through the Humboldt Research Fellowship for Postdoctoral Researcher.

**Data Availability Statement:** The Digisonde data: https://ulcar.uml.edu/DIDBase/, accessed on 1 September 2020, the Kp index: https://www.gfz-potsdam.de/en/kp-index/, accessed on 1 September 2020, the GNSS TEC data and maps: ftp://cddis.gsfc.nasa.gov/gps/products/ionex/, accessed on 1 September 2020, ftp://gnss.oma.be/gnss/products/IONEX/, accessed on 1 September 2020. The ERA5 reanalysis data: https://cds.climate.copernicus.eu/cdsapp#!/dataset/reanalysis-era5-pressure-levels?tab=form, accessed on 10 September 2020.

**Acknowledgments:** The Digisonde data were obtained from the GIRO database (https://ulcar.uml.edu/DIDBase/, accessed on 1 September 2020). The Kp index was obtained from the GFZ German Research Centre for Geosciences (https://www.gfz-potsdam.de/en/kp-index/, accessed on 1 September 2020). The GNSS TEC data and maps are provided by Center for Orbit Determination in Europe (CODE), Technical University of Catalonia (UPC), and Royal Observatory of Belgium (ROB) and are available in IONEX format at servers ftp://cddis.gsfc.nasa.gov/gps/products/ionex/, accessed on 1 September 2020 and ftp://gnss.oma.be/gnss/products/IONEX/, accessed on 1 September 2020. The ERA5 reanalysis data used for SSW determination were taken from https://cds.climate.copernicus.eu/cdsapp#!/dataset/reanalysis-era5-pressure-levels?tab=form, accessed on 10 September 2020. This work was supported by the European Space Agency project VERA (VERtical coupling in Earth's atmosphere at mid and high latitudes). We sincerely thank three reviewers who significantly increased the quality of the manuscript.

**Conflicts of Interest:** The authors declare no conflict of interest.

## Appendix A. Directograms

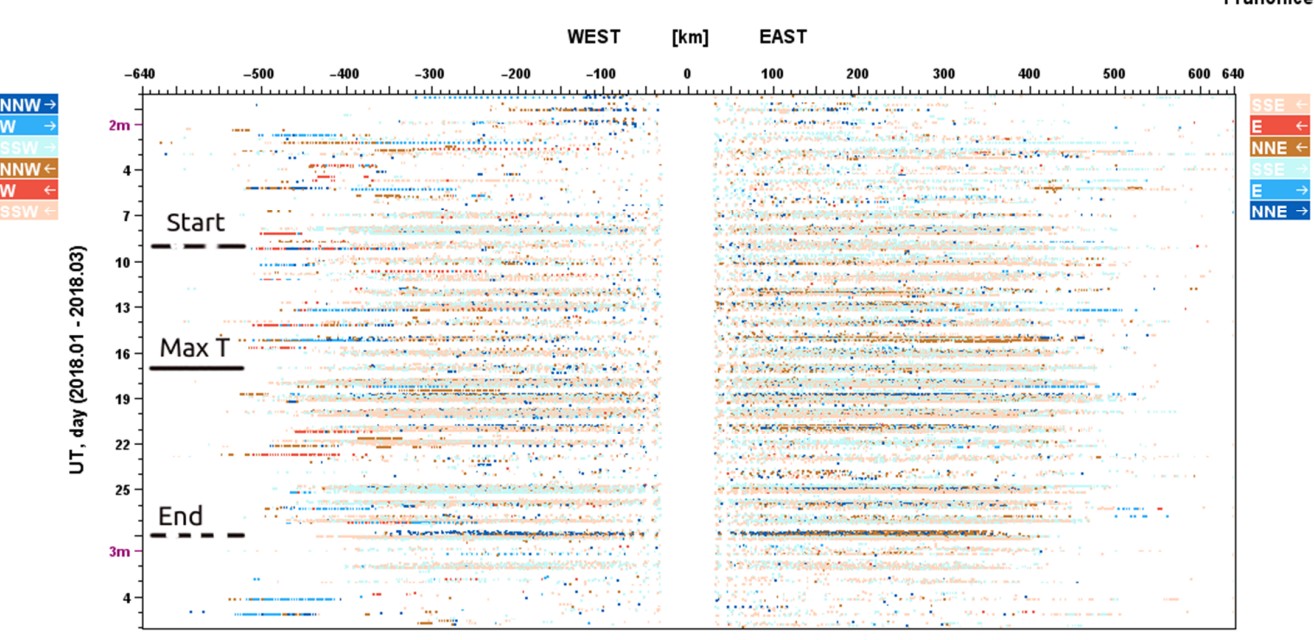

**Figure A1.** Directogram from Pruhonice station between 30 January and 6 March 2018. Increased wave activity was observed without clear connection to the SSW. Horizontal lines denote start of SSW; maximum in stratospheric temperature; and end of SSW, respectively.

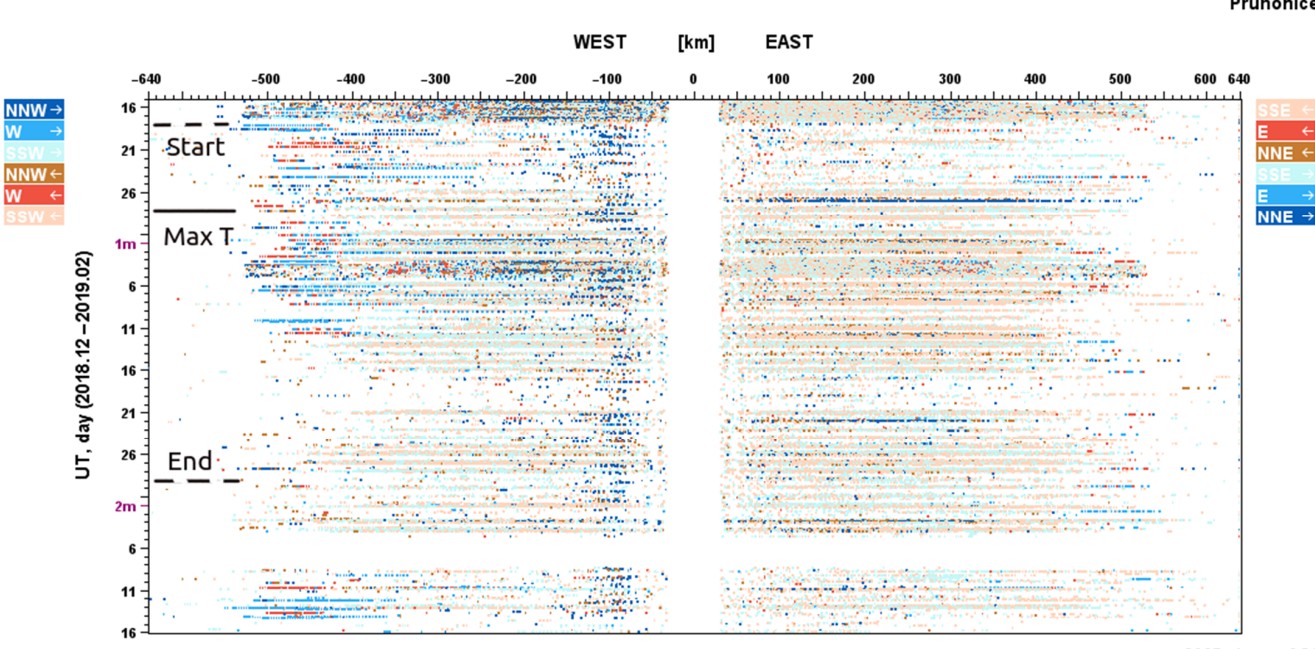

**Figure A2.** Directogram from Pruhonice station between 16 December 2018 and 16 March 2019. Increased wave activity was observed without clear connection to the SSW. Horizontal lines denote start of SSW; maximum in stratospheric temperature; and end of SSW, respectively. Horizontal lines denote start of SSW; maximum in stratospheric temperature; and end of SSW, respectively.

## Appendix B. TEC Maps

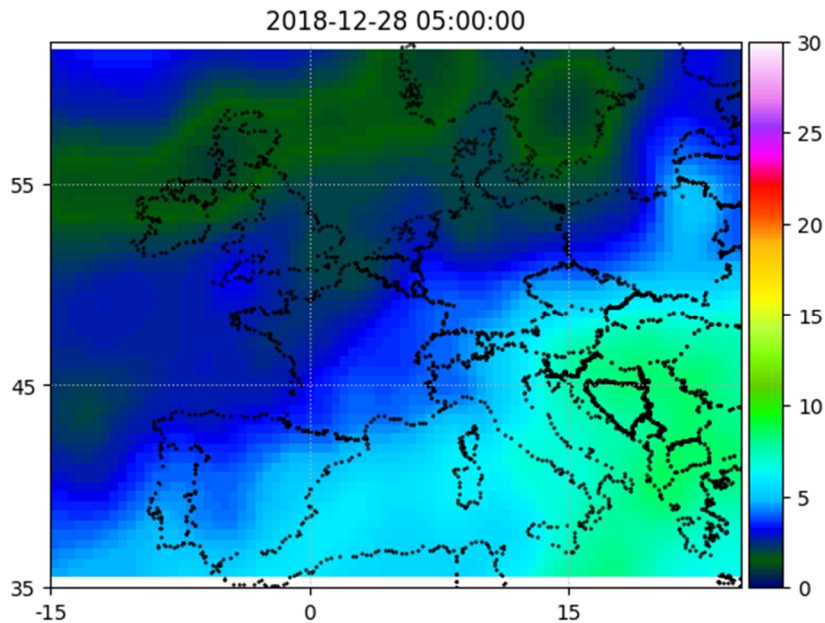

**Figure A3.** TEC map from the maximum of stratospheric temperature (Kp = 4).

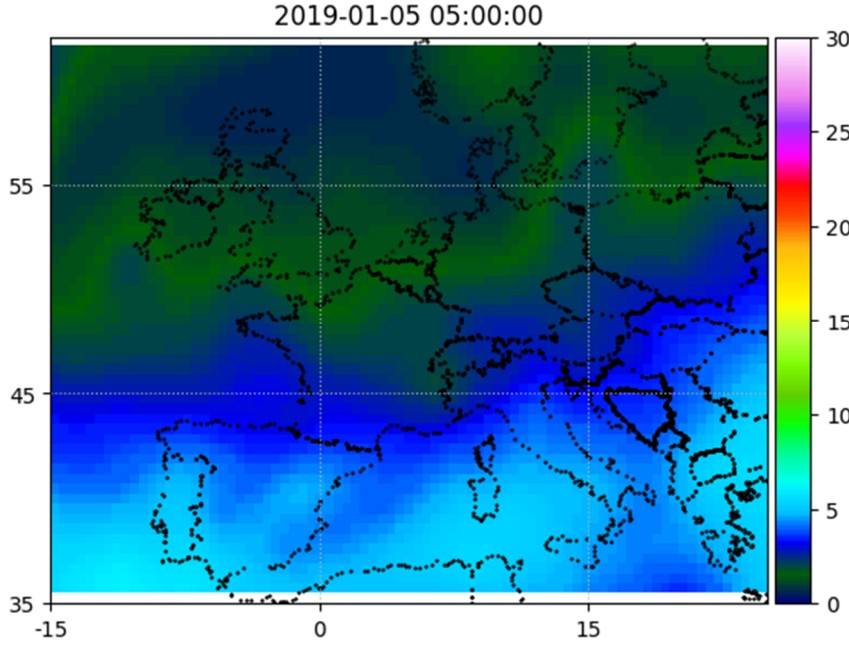

**Figure A4.** TEC map from the day of increased geomagnetic activity comparable to 28 December 2018.

**Appendix C. Peak Height of E Layer**

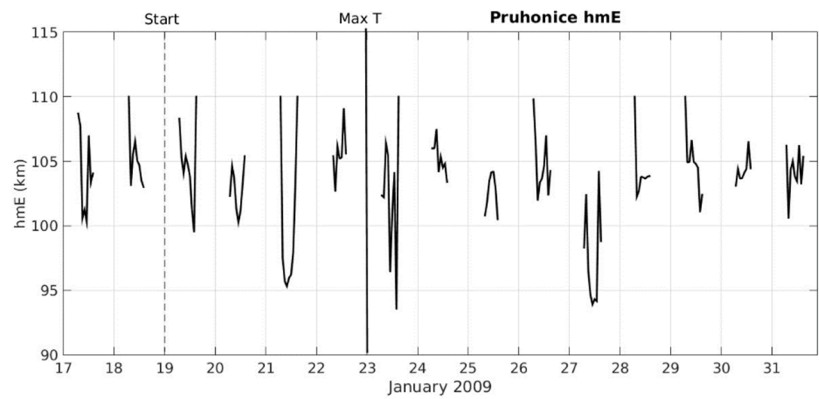

**Figure A5.** Peak height of E layer (hmE) during 2009 SSW. The dashed and solid vertical lines denote the starts of SSW and maximum in stratospheric temperature, respectively.

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
