# Peer review of "Observation of the Ionosphere in Middle Latitudes during 2009, 2018 and 2018/2019 Sudden Stratospheric Warming Events"

_atmosphere, doi:10.3390/atmos12050602_

Round 1
Reviewer 1 Report
Review of "Observation of the ionosphere in middle latitudes during 2009, 2018 and 2018/2019 Sudden Stratospheric Warming events" by Mošna et al.
The goal of this paper is to investigate the ionospheric response to 2009 and 2018 and 2018/2019 major SSW events. I believe the paper is close to being accepted for publication in MDPI Atmosphere in its present form. In my opinion, some points should be considered and clarified before publication (please see attachment).

Author Response
Dear reviewer, on behalf of all co-authors we thank you for your help and valuable comments. We answered all the questions raised. Please see the attached document for detailed answers.

Reviewer 2 Report
The paper presents the results of the analysis of three major SSWs from years of deep minima of solar activity with the aim of estimating the role of the lower atmosphere forcing on the midlatitudinal ionosphere.
The authors analyzed 3 SSW events in detail, using data from ionosondes in Europe, TEC data, and modeling in the time periods using the WACCM-X model.
This topic is undoubtedly relevant for the study of atmospheric-ionospheric coupling and the influence of sources in the lower atmosphere on the thermosphere and ionosphere.
Minor Comments:
- Line 18. “of the neutral atmosphere on the ionosphere”. The ionosphere is the part of the upper atmosphere with charged components. Perhaps, better write " ….were studied to evaluate this effect on the thermosphere and the ionosphere.”
- Line 137. add an open bracket 37].
- Line 205. Why didn't analyze the dynamics of Es?
- Fig.8, panel a. “Average over Europe”. I think that the title omitted “TEC” - “Average TEC over Europe”.
- Figures 8, 10, and 11 lack a discussion of the difference in the results between average TEC over Europe modeled TEC values over Europe and average ROTI. Why the effect of geomagnetic events in the ROTI is present, but the SSW effect is not? If possible, discuss this point in the text in more detail.
- The authors in the text indicate that “The ionospheric wave activity deduced from the directogram in Pruhonice is pronounced mainly during the period of increased temperature, and onset of zonal wind reversal.” (lines 497-499). What is the type of wave activity? Could the authors write a little more about this?
- Lines 509-511. “We observed a slight decrease in hmE corresponding to the 2009 SSW temperature maximum as well as zonal wind reversal maximum compared to background values.” I didn't see any figures in the text confirming this. Perhaps you should add a figure that will make this statement more understandable.
Undoubtedly the article “Observation of the ionosphere in middle latitudes during 2009, 2018 and 2018/2019 Sudden Stratospheric Warming events” by Zbyšek Mošna et al. may be recommended for publication after minor revision.
Author Response

(The authors gave the same response as above.)

Reviewer 3 Report
This paper studied three SSW events and evaluated the ionospheric effect in the midlatitude by using ionosonde and TEC results. This paper is interesting in exploring the dynamics and characteristics at midlatitude ionosphere during SSW though some explanations and descriptions are seemed to be inadequate. The following issues should be addressed before this reviewer can recommend the paper for publication.
Detailed Comments:
- The scientific novelty of this study should be emphasized. As the authors indicated, these three SSW events have been extensively investigated by using observations and models at midlatitude. The authors need to clearly express the novel point of their research.
- The description of the positive ionospheric storms in the last paragraph of the introduction is too simple. The mechanisms for the positive ionospheric storms are complicated, which involves penetration electric field, disturbance dynamo electric field, as well as forcing from thermospheric neutral wind and TAD/TIDs. The authors are suggested to elaborate on these aspects and cite necessary references.
- Line 432: the author said, “the situation was…influenced by minor-to-moderate geomagnetic storm (Kp <= 4+)”. This is a geomagnetic active condition but not the storm. Base on the definition given by the space weather prediction center, Kp=5 marks a minor storm.
Minor Comments:
- Zonal mean zonal wind à mean zonal wind
- Westerly and easterly à western and eastern
- Figure 3: there was no dashed and solid line to denote SSW
Author Response

(The authors gave the same response as above.)

Round 2
Reviewer 3 Report
The authors have addressed the previous comments. The paper can be published in the current form.